# MODALITY COMPLEMENTARITY: TOWARDS UNDERSTANDING MULTI-MODAL ROBUSTNESS

## ABSTRACT

Along with the success of multi-modal learning, the robustness of multi-modal learning is receiving attention due to real-world safety concerns. Multi-modal models are anticipated to be more robust due to the possible redundancy between modalities. However, some empirical results have offered contradictory conclusions. In this paper, we point out an essential factor that causes this discrepancy: The difference in the amount of modality-wise complementary information. We provide an information-theoretical analysis of how the modality complementarity affects the multi-modal robustness. Based on the analysis, we design a metric for quantifying how complementary the modalities are to others and propose an effective pipeline to calculate our metric. Experiments on carefully-designed synthetic data verify our theory. Further, we apply our metric to real-world multi-modal datasets and reveal their property. To our best knowledge, we are the first to identify modality complementarity as an important factor affecting multi-modal robustness.

## 1 INTRODUCTION

Recently, deep neural networks have proved successful in various areas, such as image recognition (He et al., 2015; Krizhevsky et al., 2012), speech recognition (Chorowski et al., 2015) and neural machine translation (Wu et al., 2016). The revolution is also happening in multi-modal research, *e.g.* RGB-D semantic segmentation (Wang et al., 2016), audio-visual learning (Zhao et al., 2018), and visual question answering (Antol et al., 2015). Intuitively, multi-modal models are anticipated to be more robust due to the potential redundancy between modalities. When one of the modalities is corrupted, others can compensate for the loss. This intuition is supported by both psychological studies of the human perception system (Sumby & Pollack, 1954) and deep learning practices (Zhang et al., 2019b; Qian et al., 2021; Wang et al., 2020).

However, some recent studies cast doubt on this belief. From a theoretical perspective, the multi-modal models usually have a larger input dimension than uni-modal models, and the increase of input dimensions significantly degrades model robustness (Ford et al., 2019; Simon-Gabriel et al., 2019). From an empirical view, some experiments suggest that multi-modal integration may be more vulnerable to attacks or corruptions than uni-modal models (Yu et al., 2020; Tian & Xu, 2021; Ma et al., 2022).

What causes this contradiction in multi-modal robustness? We notice that the conclusions above are drawn under assorted multi-modal task settings ranging from action classification to question answering, which vary in the presence and type of modality interconnections (Liang et al., 2022). Therefore, a question arises naturally:

*What aspects of modality interconnection affect the multi-modal robustness?*

We hypothesize that the *complementarity of modalities* plays an essential role. If the complementary part of each modality is negligible, the corruption of one modality would not severely damage the model performance. Otherwise, the multi-modal model could perform even worse than a uni-modal model. For the visual question answering task, the two modalities are highly complementary: Only perceiving either the question or the image could not lead to an ideal answer (Agrawal et al., 2018). For the action classification task, the RGB and optical flow are less complementary since each of them can suggest a roughly correct answer (Feichtenhofer et al., 2016b).

To validate the above hypothesis, we first demonstrate the key role of modality complementarity to model robustness through theoretical analysis. Following previous work (Tsai et al., 2020; Sun et al., 2020; Sridharan & Kakade, 2008; Tosh et al., 2021), we use an information-theoretical framework for multi-modal learning and study how the complementary information affects robustness under missing and noisy modality settings. Based on the analysis, we design a novel metric and a practical calculation pipeline built on Mutual Information Neural Estimator (MINE) (Belghazi et al., 2018) to quantify the complementarity of modalities in multi-modal datasets.

With the specially designed metric and pipeline on hand, we verify our theory and the effectiveness of our proposed metric on synthetic data and a carefully-designed toy dataset AAV-MNIST. The results are consistent with the model robustness in modality missing, noisy modality, and adversarial attack settings on the datasets we test on. Then we apply our metric to real-world multi-modal datasets to further investigate the modality complementarity in different settings. To our best knowledge, we are the first to identify and prove the important role of modality complementarity in multi-modal robustness. Hence, for future research, we recommend that researchers consider the modality complementarity as a control variable for a fairer comparison of multi-modal robustness.

The main contributions are highlighted as follows:

- We point out the effect of modality complementarity on multi-modal model robustness through information-theoretical analysis.
- We propose a dataset-wise metric to qualitatively evaluate how complementary the modalities are in each multi-modal dataset, and also design a pipeline for computing the metric in real-world datasets.
- We create a synthetic dataset and a toy dataset (AAV-MNIST) to test our metric and pipeline. These datasets cover various complementary situations of different modalities and are used to verify the effectiveness of our pipeline.
- We further reveal the modality complementarity and its relationship with model robustness in real-world multi-modal datasets, which could lead to a less biased comparison for multi-modal robustness.

## 2 RELATED WORK

**Multi-modal learning.** Various multi-modal learning tasks and models are proposed in recent years (Baltrusaitis et al., 2017; Liang et al., 2021), such as multi-modal reasoning (Yi et al., 2019; Johnson et al., 2016), cross-modal retrieval (Gu et al., 2017; Radford et al., 2021), and cross-modal translation (Ramesh et al., 2021). Among these settings, we mainly focus on the supervised multi-modal classification setting. The theoretical understanding of multi-modal learning is relatively under-explored, with (Huang et al., 2021) deriving generalization error bounds and (Sun et al., 2020) comparing with the Bayesian posterior classifiers. A concept close to multi-modal learning is the multi-view learning (Xu et al., 2013). The theory of multi-view learning has long been studied both theoretically (Zhang et al., 2019a; Tosh et al., 2021) and empirically (Sindhwani et al., 2005; Ding et al., 2021; Amini et al., 2009; Tian et al., 2019). Earlier work (Kakade & Foster, 2007; Sridharan & Kakade, 2008) proposes the multi-view assumption: Each modality suffices to predict the label. Recently, many multi-view analyses adopted this assumption (Han et al., 2021; Tsai et al., 2020; Lin et al., 2021; Federici et al., 2020; Lin et al., 2022). However, as pointed out by (Huang et al., 2021; 2022), this might not hold in the multi-modal learning setting.

**Model robustness.** Model robustness under data missing (Ramoni & Sebastiani, 2001), random corruption (Hendrycks & Dietterich, 2019), and adversarial attacks (Madry et al., 2017) is constantly been concerned in consideration of real-world safety issues. For uni-modal models, several methods are proposed to strengthen model robustness (Papernot et al., 2015; Huang et al., 2015; Meng & Chen, 2017). For multi-modal models, some papers regard the use of multi-modality as a way to improve robustness (Zhang et al., 2019b; Qian et al., 2021; Wang et al., 2020), while others continue to improve multi-modal models' robustness by designing new network architectures and fusion methods (Kim & Ghosh, 2019a; Tsai et al., 2018; Yang et al., 2021) and training routines (Eitel et al., 2015; Ma et al., 2021). When dealing with known missing patterns, researchers explore additional ways: data imputation through available modalities or views (Tran et al., 2017; Lin et al., 2021), or training different models for different availability of modalities (Yuan et al., 2012). Our analysis

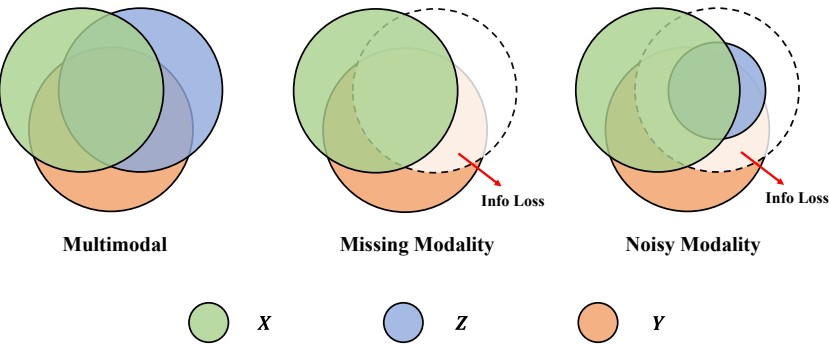

Figure 1: Illustration of relationships between the inputs and the target of a multi-modal task in different cases. $X$ and $Z$ are random variables representing the input of two modalities. $Y$ is the target we would like to infer. The Info Loss refers to the loss of $Y$-relevant information provided by inputs, which is caused by missing or corruption of modality $Z$.

points out the upper bound of these methods. Apart from improving robustness, another line of work aims to analyze or estimate the robustness of existing uni-modal methods (Cohen et al., 2019; Carlini et al., 2019; Mahmood et al., 2021) and multi-modal methods (Yu et al., 2020; Tian & Xu, 2021; Ma et al., 2022; Rosenberg et al., 2021; Li et al., 2020). We in this work analyze one factor of multi-modal model robustness both theoretically and empirically.

**Mutual information in deep learning.** Mutual information is tightly related to deep learning through multiple ways, including information bottleneck method (Tishby et al., 2000), analysis of learning methods (Wu & Verdu, 2012; Tsai et al., 2020; Shwartz-Ziv & Tishby, 2017), and new learning methods based on mutual information (Hjelm et al., 2018; Bachman et al., 2019; Sun et al., 2020). On the other hand, learning methods help to estimate the amount of mutual information. Representative work includes Mutual Information Neural Estimator (MINE) (Belghazi et al., 2018), CPC (van den Oord et al., 2018), DIM (Hjelm et al., 2018), and DoE estimator (McAllester & Stratos, 2018). We apply the MINE to our calculation pipeline for its simplicity and effectiveness.

## 3 THEORETICAL ANALYSIS

In this section, we first build an information-theoretical framework for multi-modal learning and show the impact of complementary information to model robustness in modality missing and single noisy source cases, which are commonly studied in previous work (Kim & Ghosh, 2019a; Tian & Xu, 2021) and widely encountered in practice, e.g., some sensors are broken, prone to noise (e.g. cameras in foggy environments), or expensive to use(e.g. X-ray data for medical analysis). An illustration of these cases is plotted in Figure 1.

### 3.1 PRELIMINARIES

**Notations.** We use $H(A)$ to represent the entropy of a random variable $A$, $H(A|B)$ for the conditional entropy given another variable $B$, $I(A;B)$ for the mutual information between random variable $A$ and $B$, $I(A;B|C)$ for the conditional mutual information conditioned on random variable $C$, and $I(A;B;C)$ for the interaction information (i.e., mutual information of three variables, possibly a negative value).

**Multi-modal learning formulation.** We adopt the formulation for multi-modal learning problem proposed in (Huang et al., 2021) Denote the $M$-modality input space as $\mathcal{X} = \mathcal{X}_1 \times \mathcal{X}_2 \times \ldots \mathcal{X}_M$ and the target space as $\mathcal{Y}$. Each data point $(X_1, X_2, \ldots, X_M, Y)$ is sampled from an unknown distribution on $\mathcal{X} \times \mathcal{Y}$. Our goal is that, based on the random input variables $X_1, X_2, \ldots, X_M$ from $M$ modalities, we would like to infer the target $Y$. In classification tasks, $Y$ is a discrete random variable, while in regression tasks $Y$ is continuous. For instance, considering audio-visual action recognition (Gao et al., 2019; Feichtenhofer et al., 2016a), let $X_1$ be the audio part and $X_2$ be the frames of a video clip, and we want to infer the label $Y$, i.e., what kind of action is performed in the clip. In the subsequent analysis, we will focus on the common case $M = 2$ (Feichtenhofer et al., 2016a) for simplicity, and we denote the two modalities as $X \in \mathcal{X}$ and $Z \in \mathcal{Z}$ respectively. Notice

that our analysis and results can be extended to cases with more than two modalities at the expense of notations.

**Complementary information.** Now we define the *complementary information* in the following, which is essential through our theoretical analysis.

**Definition 1** (complementary information). *For input variables $X$, $Z$ and the target $Y$, define the complementary information provided by $X$, $Z$ as follows*

$$\Gamma_{X,Y} = I(X; Y|Z)$$
$$\Gamma_{Z,Y} = I(Z; Y|X)$$

*When the target is clear from the context, we omit the $Y$ in the subscript.*

Mathematically, $I(X; Y|Z)$ represents the information in the target $Y$ that is accessible for $X$ but not predictable for $Z$. Thus $\Gamma_X$ can characterize the unique label information owned by modality $X$, and similarly for $\Gamma_Z$. Hence $\Gamma_X$ together with $\Gamma_Z$ can determine the complementarity of modality $X$ and $Z$. Clearly, larger $\Gamma_X$ and $\Gamma_Z$ imply higher complementary information content.

From the standard derivation in information theory, we can obtain the following relation:

$$I(X, Z; Y) = \Gamma_X + \Gamma_Z + I(X; Y; Z) \tag{1}$$

Previous theoretical analyses of multi-view learning Sridharan & Kakade (2008); Xu et al. (2013); Tosh et al. (2021) usually adopt the multi-view assumption that each view is redundant in terms of predicting the target, i.e. $\Gamma_X$ and $\Gamma_Z$ are both small. However, this assumption does not always hold in the multi-modal learning setting Antol et al. (2015). In the following subsections, we will show how the complementary information $\Gamma_X$ and $\Gamma_Z$ affect the model robustness in missing modality and noise settings. Motivated by this theoretical observation, we will propose a metric to evaluate the modality complementarity and a pipeline for calculation in Section 4.

**Bayes error rate.** We introduce the Bayes error rate Fukunaga & Hummels (1987) to measure the model performance, which is the lowest possible error for any arbitrary classifier or predictor from the multiple modalities to infer the target. Formally, given two modalities $X$ and $Z$, the multi-modal Bayes errors for classification $P_{e_c}$ and regression $P_{e_r}$ are defined as follows:

$$P_{e_c} := \mathbb{E}_{x,z \sim P_{X,Z}}[1 - \max_{y \in Y} P(Y = y|x, z)]$$

$$P_{e_r} := \mathbb{E}_{x,z,y \sim P_{X,Z,Y}}[(y - \mathbb{E}[Y|x, z])^2]$$

The Bayes error rate helps us focus on the interconnection among modalities $X$, $Z$ and target $Y$ in each multi-modal task and omit other factors' effects on model robustness, e.g. dataset size, training routines, and network architectures.

## 3.2 Missing Modality

We first consider the missing modality scenario and assume modality $Z$ is missing w.l.o.g.. Then the Bayes error rates for missing modality, denoted as $P_{e_c}^{\text{Miss}}$ and $P_{e_r}^{\text{Miss}}$ become

$$P_{e_c}^{\text{Miss}} = \mathbb{E}_{x \sim P_X}[1 - \max_{y \in Y} P(Y = y|x)]$$

$$P_{e_r}^{\text{Miss}} = \mathbb{E}_{x,y \sim P_{X,Y}}[(y - \mathbb{E}[Y|x])^2].$$

Now we establish the following theoretical guarantees to quantify differences between the Bayes error rate of multi-modal and missing-modality.

**Theorem 3.1.** *For random variables $X, Z$ and discrete random variable $Y$, we have*

$$\frac{H(Y|X, Z) - \log 2}{\log |Y|} \le P_{e_c} \le 1 - \exp(-H(Y|X, Z)) \tag{2}$$

$$\frac{H(Y|X, Z) + \Gamma_Z - \log 2}{\log |Y|} \le P_{e_c}^{Miss} \le 1 - \exp(-H(Y|X, Z) - \Gamma_Z) \tag{3}$$

*For continuous random variable $Y$, if we further assume that $Y$ takes value in $[-1, 1]$, then we have*

$$P_{e_r}^{Miss} - P_{e_r} \le \frac{1}{2}\Gamma_Z \tag{4}$$

$\square$

*Remark* 1. The **gap** between $P_{e_c}^{\text{Miss}}$ (best model performance in modality missing setting) and $P_{e_c}$ (best model performance in normal setting) reflects the best model robustness against modality missing. For the classification task, when $\Gamma_Z = 0$, i.e., there is no complementary information of $Z$, the information from $Z$ can be covered by the information from $X$ for predicting $Y$. In this case, the $P_{e_c}^{\text{Miss}}$ shares the same lower and upper bound with $P_{e_c}$, so the performance of the best model would not be affected by modality missing. As the $\Gamma_Z$ increases, the bounds for $P_{e_c}^{\text{Miss}}$ rise, while the bounds for $P_{e_c}$ is fixed, indicating that the best model performance drops under modality missing, i.e. the robustness decays. Considering the extreme case when $\Gamma_Z$ is large enough, the lower bound of $P_{e_c}^{\text{Miss}}$ is greater than the upper bound of $P_{e_c}$, so the missing modality performance is provably worse than normal performance.

*Remark* 2. For the regression task, the closer $P_{e_r}^{\text{Miss}}$ and $P_{e_r}$ are, the robust the best model is. From the result above, the gap between two Bayes optimal predictors is bounded above by the complementary information, hence increased by $\Gamma_Z$. So the model robustness under modality missing is worsened along with the increase of $\Gamma_Z$.

### 3.3 SINGLE NOISY MODALITY

The modality corrupted by noise is another situation that we often encounter in practice, e.g., the foggy weather results in noisy RGB images in autonomous driving. In this section, we study the case that one of the modalities has additional noise, which can be easily extended to the case that all modalities are noisy at the expense of notations. Formally, we consider that Gaussian noise $N$ is added to the input modality $Z$ (Zheng et al., 2016; Kim & Ghosh, 2019b). We use $R_N(Z) = Z + N$ to denote the modality $Z$ after adding Gaussian noise. By (Cover, 1999) we can obtain the following characterization for the mutual information between $Z$ and $R_N(Z)$

**Proposition 1.** *If $Z, N \in \mathbb{R}$, assuming that $0 < \mathbb{E}[Z^2] \le p_Z$, $N \sim \mathcal{N}(0, \sigma)$, and $N$ is independent of $Z$, then we have*

$$I(Z; R_N(Z)) \le \frac{1}{2} \log(1 + \frac{p_Z}{\sigma}) \tag{5}$$

*Remark* 3. When the noise is heavy, i.e., the $\sigma$ is large, the upper bound of $I(Z; R_N(Z))$ decays, indicating that it is harder to recover $Z$ from $R_N(Z)$ and thus harder to infer $Y$ from $R_N(Z)$. When the noise becomes very heavy, $I(Z; R_N(Z))$ will be near zero and $R_N(Z)$ is close to pure Gaussian noise, as if the modality $Z$ is missing, which suits our intuition. In this extreme case, we can refer to the analysis in section 3.2.

In this setting, the Bayes error rate for classification denoted as $P_{e_c}^{\text{No}}$ can be written as:

$$P_{e_c}^{\text{No}} = \mathbb{E}_{x,z \sim P_{X,Z}}[1 - \max_{y \in Y} P(Y = y|x, R_N(z))]$$

Then we can provide the lower bound for $P_{e_c}^{\text{No}}$.

**Theorem 3.2.** *For random variables $X, Y, Z, N$, if $\mathbb{E}[Z^2] \le p_Z$, $N \sim \mathcal{N}(0, \sigma)$, then*

$$P_{e_c}^{No} \ge \frac{H(Y|X, Z) + \Gamma_Z + I(X; Y; R_N(Z)) - \frac{1}{2} \log(4 + \frac{4p_Z}{\sigma})}{\log |Y|} \tag{6}$$

$\square$

*Remark* 4. Similar to the analysis in modality missing setting, the gap between $P_{e_c}^{\text{No}}$ (best model performance in noisy modality setting) and $P_{e_c}$ (best model performance in normal setting) reflects the best model robustness against noisy modality. For the classification task, the lower bound of $P_{e_c}^{\text{No}}$ increases as $\Gamma_Z$ or $\sigma$ increases. Since the bounds of $P_{e_c}$ are fixed, the gap between $P_{e_c}^{\text{No}}$ and $P_{e_c}$ becomes larger, and the model robustness under noisy setting is worse. Therefore, if the $\Gamma_Z$ is larger, the best predictor become more vulnerable to the added noise.

## 4 METRIC

In this section, we propose a dataset-wise metric based on the complementary information to quantify the modality complementarity. We also bring our metric to practical use by leveraging the existing mutual information estimator, Mutual Information Neural Estimator (MINE) (Belghazi et al., 2018).

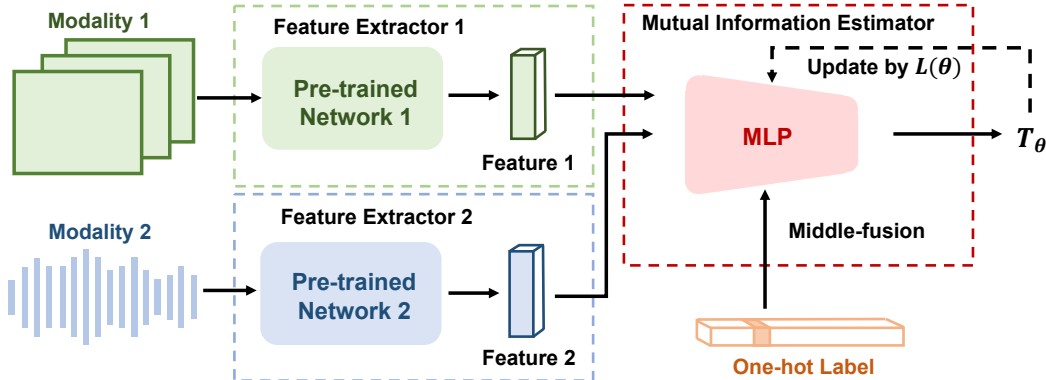

Figure 2: Pipeline to calculate the metric: First extract the features of the input data from two modalities by pre-trained models. Then apply the MINE to estimate the mutual information $I(Z; Y, X)$, $I(X; Y, Z)$, or $I(X; Z)$.

## 4.1 METRIC DESIGN

From the above analysis, it is natural to use $\Gamma_X + \Gamma_Z$ as the metric since they represent how much complementary information the modalities $X$ and $Z$ can provide exclusively about the target $Y$.

However, $\Gamma_X + \Gamma_Z$ is not enough for comparing among datasets. According to equation 1, the same amount of $\Gamma_X + \Gamma_Z$ could indicate different situations if the total information $I(X, Z; Y)$ is different. Therefore, to make the metric comparable among datasets, we need to perform normalization by dividing it with $I(X, Z; Y)$, written as

$$\frac{\Gamma_X + \Gamma_Z}{I(X, Z; Y)}$$

Now, our metric becomes the "proportion" of $\Gamma_X + \Gamma_Z$ in $I(X, Z; Y)$. When our metric is large, the modalities are more complementary to each other and more indispensable for the task. Note that this "proportion" could be greater than 1 because $I(X; Y; Z) = I(X, Z; Y) - \Gamma_X - \Gamma_Z$ may be negative. This happens when $Z$ (or $X$) greatly increases the correlation strength between $X$ (or $Z$) and $Y$. Without $Z$ (or $X$), the other modality becomes nearly uncorrelated with the target $Y$. Hence, when the metric is greater than 1, it can still reflect the modality complementarity and reveals more about the interconnection between the modalities and the target.

## 4.2 CALCULATION

Now we consider how to calculate our metric. $\Gamma_X$ and $\Gamma_Z$ are in the form of conditional mutual information and could not be computed directly. We notice that

$$\Gamma_Z = I(Z; Y|X) = I(Z; Y, X) - I(Z; X)$$
$$\Gamma_X = I(X; Y|Z) = I(X; Y, Z) - I(X; Z)$$

So we transform the metric into

$$\frac{\Gamma_X + \Gamma_Z}{I(X, Z; Y)} = \frac{I(X; Y, Z) + I(Z; Y, X) - 2I(X; Z)}{I(X, Z; Y)}$$

Additionally, considering that most real-world datasets roughly satisfy the realizability assumption, i.e., there exists a function in the hypothesis space that can predict $Y$ given $X$ and $Z$ with zero population risk, we could approximate $I(X, Z; Y) = H(Y) - H(Y|X, Z)$ with $H(Y)$ because the second term is close to zero. $H(Y)$ is easier to compute given the distribution of $Y$, especially when we focus on the classification task with discrete labels.

For each mutual information term with the form $I(A; B)$, we design a two-phase pipeline for computation (See Figure 2):

- In the first phase, we reduce the dimension of the high-dimensional input $A$ and $B$ to accelerate the computation by pre-trained feature extractors. The pre-trained models are shared among the calculation of all three terms.

- In the second phase, we use the extracted features as inputs for MINE (Belghazi et al., 2018) to compute the mutual information. Specifically, we calculate the value through optimization converging to a lower bound of the mutual information. For each iteration, we sample an m-sample batch $\{(\mathbf{a}^{(i)}, \mathbf{b}^{(i)})\}_{i=1}^{m}$ from the joint distribution $P(A, B)$ and an m-sample batch $\{\mathbf{b}'^{(i)}\}_{i=1}^{m}$ from the marginal distributions $P(B)$. Denote the estimator network as $T$ and its parameters as $\theta$. We evaluate the lower bound $L$ as follows and the moving average of gradients of $L(\theta)$ for updating the network parameters.

$$L(\theta) = \frac{1}{m} \sum_{i=1}^{m} T_\theta(\mathbf{a}^{(i)}, \mathbf{b}^{(i)}) - \log(\frac{1}{m} \sum_{i=1}^{m} \exp(T_\theta(\mathbf{a}^{(i)}, \mathbf{b}'^{(i)})))$$

  We adjust the original MINE by adding the following trick: The calculation of $I(X; Y, Z)$ and $I(Z; Y, X)$ involve the target $Y$, so we concatenate the one-hot encoding label with the extracted feature in a middle-fusion fashion and ensure that the estimator network $T$ could combine the two information sources. For more details, please see the supplementary material.

We believe that modality complementarity is crucial to the analysis of multi-modal robustness. Without controlling this factor, we cannot fairly compare experimental results on various multi-modal datasets, and thus we cannot derive a universal conclusion on multi-modal robustness. By calculating our metric on multi-modal datasets, we will better understand their difference in modality complementarity, leading to less biased comparisons and conclusions.

## 5 EXPERIMENTS

We conduct experiments to verify the validity of our analysis and the effectiveness of our pipeline. We first introduce the training and testing settings and then show the results on the synthetic dataset, Additive AV-MNIST dataset, and real-world datasets. Unless otherwise specified, the missing/noise/adversarial robustness mentioned in the following subsections refers to the average accuracy under two sources of missing/noise/adversarial attack, divided by the model accuracy in the clean setting. For more detailed settings and results, please see the Appendix B and C.

**Training setting.** We use an MLP as the estimator. For different datasets, the structure of the MLP varies to match the input size. We train the estimator on the training set since in reality we only have access to it, and we assume that the validation set is i.i.d. sampled from the same distribution as the training set. For the pre-trained feature extractors, we use Resnet18 (He et al., 2015) and AudioNet (1-D CNN) (Tian & Xu, 2021) for Kinetics-Sounds and AVE, LeNet5 (LeCun et al., 1998) and an 2-D CNN for AAV-MNIST, and Resnet152 (He et al., 2015) and BERT (Devlin et al., 2019) for Hateful-Meme dataset. For the models tested for robustness, we use late fusion models for AVE, Kinetics-Sounds, and AV-MNIST, and we apply MMBT (Kiela et al., 2019) for Hateful-Meme dataset.

**Test setting.** We test the model robustness under two settings discussed above: missing modality and single noisy modality. We also explore the model robustness under adversarial attack. For the missing image or audio, we substitute them with the average of all inputs in the training set. For the missing text, we use a blank sentence <SOS><EOS> as the input. Note that the inputs are all scaled to the range $[-1, 1]$ (spectrogram) or $[0, 1]$ (image). For noisy image and audio, we add a Gaussian noise $N \sim \mathcal{N}(0, 0.5)$ to each dimension. For noisy text, we replace each word by a random word with a probability 0.5. For adversarial attack on image and audio, we use FGSM (Goodfellow et al., 2014) with step size $\epsilon = 0.03$. We use the results of missing text for adversarial text.

### 5.1 SYNTHETIC DATASET

We first test our analysis on a well-designed synthetic dataset since we can adjust the degree of its modality complementarity. Inspired by previous work (Hessel & Lee, 2020) and (Huang et al., 2021), we generate a set of synthetic data $(x, z, y)$. First, we sample random projection $P_X \in \mathbb{R}^{d_1 \times d}$, $P_Z \in \mathbb{R}^{d_2 \times d}$, and $P \in \mathbb{R}^{d \times d}$ from a uniform distribution $U(-0.5, 0.5)$. Then we repeat following steps:[1]

---

[1]We sample 5000 data points with 80/20 train/val split and $\langle d, d_1, d_2, \delta \rangle = \langle 50, 200, 100, 0.25 \rangle$.

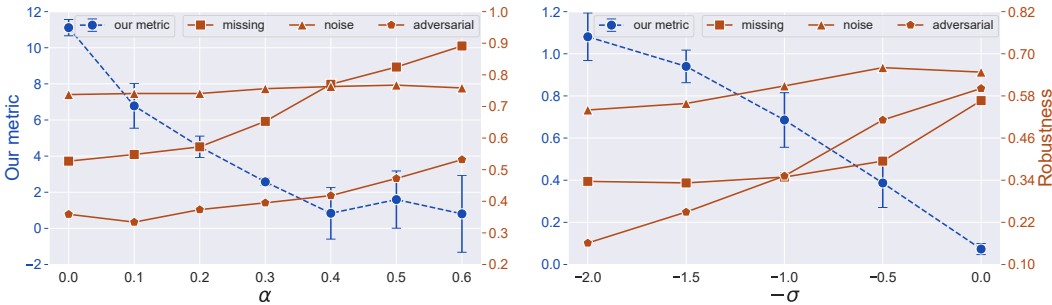

Figure 3: Two line plots showing the mean value of estimated metric (blue line) with error bars (standard variances) of three independent repeated experiments and tested robustness (orange line) on the synthetic dataset (left) and AAV-MNIST dataset (right). The x-axis is the parameter used in data generation. For the synthetic dataset, we plot $\alpha$. For AAV-MNIST, we plot $-\sigma$ for unity. As the overlap of two modalities becomes larger, they are less complementary and our metric correspondingly goes down. Meanwhile, the tested robustness increases in all three settings. The variance and trend of each mutual information term can be found in the supplementary material.

Step 1. Sample $x, z \in \mathbb{R}^d \sim \mathcal{N}(0, 1)$.

Step 2. Set $z \leftarrow (1 - \alpha)z + \alpha x$; Then do projection $z \leftarrow Pz$.

Step 3. Normalize $x, z$ to unit length; if $|x \cdot z| \leq \delta$, return to the Step 2.

Step 4. Generate the label $y$: If $x \cdot z > 0$, then $y = 1$; else $y = 0$. Return the tuple $(P_X x, P_Z z, y)$.

The $\alpha$ used in data generation controls the overlap between the two modalities $X, Z$. When $\alpha = 0$, the modalities are independent and complementary in predicting the label $Y$. When $\alpha = 1$, they are redundant for prediction. Viewing the synthetic dataset with different $\alpha$ as different datasets, we calculate our metric using the pipeline in section 4 and test the robustness of simple two-layer perceptron neural networks [2] trained on these datasets. The results are shown in the plot 3. In each dataset, our pipeline can estimate the proposed metric and quantify the complementarity of the two modalities. Further, the model robustness decreases as the complementarity increases, which verifies our analysis.

## 5.2 Additive AV-MNIST

To show that our pipeline can effectively estimate the modality complementarity of more complex datasets, we further design a toy dataset named Additive AV-MNIST (AAV-MNIST) adapted from the AV-MNIST dataset (Vielzeuf et al., 2018). The modality complementarity can be controlled by a parameter $\sigma$ in the data generation process. Below, we show how to generate AAV-MNIST dataset from the original AV-MNIST dataset. The following steps are repeated for every image $i$ in AV-MNIST:

Step 1. Let $x$ be the label of $i$. Sample $\delta \in \mathbb{R} \sim \mathcal{N}(0, \sigma)$ and round $\delta$ to the nearest integer.

Step 2. Set $y \leftarrow (x + \delta) \mod 10$. Uniformly sample a spectrogram $s$ from all spectrograms in AV-MNIST with label $y$.

Step 3. Calculate the new label $t \leftarrow (x + y)/2$. Round $t$ to the nearest integer. Return the tuple $(i, s, t)$.

The AAV-MNIST dataset is an extension of AV-MNIST dataset. When $\sigma = 0$, AAV-MNIST dataset is equivalent to AV-MNIST dataset where each image and its paired spectrogram represent the same number. As $\sigma$ increases, each image become less correlated with its paired spectrogram, so they become more complementary for predicting the label $t$.

We show in the plot 3 that our metric reflects the complementarity of the AAV-MNIST dataset with different $\sigma$, indicating that our pipeline is effective in more complex settings beyond the synthetic

---

[2]Each achieves accuracy $> 96\%$ on corresponding validation sets. The neural network structure and more training settings can be found in the supplementary materials.

| Dataset | Our metric | Clean | Missing | Noisy | Adversarial |
|---|---|---|---|---|---|
| AAV-MNIST($\sigma = 2.0$) | 0.9212 | 0.6435 | 0.3368 | 0.5399 | 0.1612 |
| Hateful-Meme | 0.2403 | 0.3249 | 0.1005 | 0.5171 | 0.3144 |
| AV-MNIST | 0.0490 | 0.9969 | 0.5666 | 0.6478 | 0.6012 |
| Kinetics-Sounds | 0.0455 | 0.6387 | 0.5540 | 0.6098 | 0.2672 |
| AVE | 0.0126 | 0.7637 | 0.4838 | 0.5831 | 0.3355 |

Table 1: Our estimated metric and tested robustness of real-world datasets: Kinetics-Sounds, AVE, AV-MNIST, and Hateful-Meme. Since the Hateful-Meme Challenge is a binary classification task, we use F1 score for evaluation instead of accuracy. We also provide results in clean setting for reference.

dataset. Further, the robustness in the three settings verifies our conclusion that with other conditions unchanged, the more complementary the modalities are, the less robust the best model will be.

## 5.3 REAL-WORLD DATASETS

Now we apply our pipeline to real-world datasets to investigate their modality complementarity. Our results on AVE (Tian et al., 2018), Kinetics-Sounds (Carreira & Zisserman, 2017; Arandjelovic & Zisserman, 2017), and Hateful-Meme dataset (Kiela et al., 2020a) are shown in the table 1. The details of these datasets are described in the Appendix B. We also list results on the AV-MNIST dataset and AAV-MNIST ($\sigma = 2.0$) for reference.

The low value in our metric of AVE, Kinetics-Sounds, and AV-MNIST indicates that they possess relatively little modality complementarity, revealing the heavy redundancy between the two modalities. On the contrary, the modalities in the Hateful-Meme dataset are more complementary. This finding suits our intuition: In the Hateful-Meme dataset, altering the paired text of an image probably changes the label (Kiela et al., 2020b). Hence, only perceiving the image would not derive the right answer. For the event classification task defined by Kinetics-Sounds or AVE, the audio and frames both lead to a rough answer.

The tested robustness demonstrates how the modality complementarity affects model robustness. The missing case affects AAV-MNIST($\sigma = 2.0$) and Hateful-Meme far more than the other three datasets. They are also more vulnerable in single source noisy case than other datasets. Hence, to compare model robustness among these datasets, we should take modality complementarity into account. For instance, we only compare robustness among datasets with a similar degree of modality complementarity, or we can normalize the results by our metric. We show an analysis of the existing measure of modality missing robustness by applying our metric in the Appendix C.3.

Furthermore, the model robustness, especially the adversarial robustness, is also affected by factors other than modality complementarity. For instance, the model adversarial robustness of AVE and Kinetics-Sounds dataset is significantly lower than that of AV-MNIST dataset. We conjecture that this is related to the number of robust features in each modality of the datasets, which requires future work to confirm.

## 6 CONCLUSIONS

In this work, we partly explain the contradiction in previous conclusions on multi-modal robustness by pointing out the importance of the modality complementarity through information-theoretical analysis and carefully-designed experiments. As a reflection of modality interconnection, our proposed metric provides a basis for better understanding various multi-modal datasets/tasks and can be used beyond analyzing multi-modal robustness.

## REPRODUCIBILITY STATEMENT

We provide the source code and configuration for the key experiments, including instructions on generating data, training the models, and evaluating the robustness. We thoroughly checked the code implementations and empirically verified the effectiveness of our method. All proofs are stated in the appendix with explanations and underlying assumptions.

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
