# OpenReview forum: "Modality Complementariness: Towards Understanding Multi-modal Robustness"
_ICLR.cc/2023/Conference — Submitted to ICLR 2023_

### Official Review · Reviewer_EDnT · 2022-10-15

**Confidence:** 4
**Clarity, Quality, Novelty And Reproducibility:** Good quality and clarity.
**Correctness:** 4
**Technical Novelty And Significance:** 3
**Empirical Novelty And Significance:** 3
**Recommendation:** 6

**Strength And Weaknesses:**

Pros:

1. This work presents a very detailed theoretical analysis of multi-modal robustness. The complementary is estimated through a litter modification on MINE with a novel information metric.
2. The organization of the paper is good in general.

Cons:

1. My main concern with this paper is its usability. To analyze the complementariness, the methods need the label information beforehand. However, as we have known the label information, one could directly train the multi-modal network and percept the robustness intuitively. I think the use of label information is an obvious drawback of this work.
2. Some works focus on the reliability of the modality [1], which could also learn the uncertainty (importance) of each modality, which is also a kind of robustness. In contrast, the proposed method could only analyze the robustness of the datasets, ignoring an important perspective, i.e., improving the representation of the network.
3. Many recent multi-view works about information theory are missing [2-5]. All these methods attempt to establish the relationship between the different modalities and the label information in an unsupervised manner, although these methods are based on some extra assumptions, e.g., mutual redundancy [3,5] or containing view-specific information [2]. Especially, [3,4] also discuss the usability of the missing modality from information theory. The paper could do a better job by citing these relevant works and clarifying the differences between this work and them.

[1] Trusted Multi-View Classification, ICLR'21

[2] Self-supervised Learning from a Multi-view Perspective, ICLR'21

[3] Dual Contrastive Prediction for Incomplete Multi-View Representation Learning, TPAMI'22

[4] COMPLETER: Incomplete Multi-view Clustering via Contrastive Prediction, CVPR'21

[5] Learning Robust Representations via Multi-View Information Bottleneck, ICLR'20

**Summary Of The Paper:**

This paper gives an information-theoretical abstraction to represent  the modality robustness through the proposed complementary metric. The deduction is based on the assumption that models are anticipated to be more robust due to the possible redundancy between modalities. The authors also do some controlled experiments and try to support their theoretical analysis.


**Summary Of The Review:**

I recommend accepting the paper (rating 6). I liked the abstraction proposed by the authors and particularly liked the way authors make connections between the complementariness with missing modality and noisy modality. Although this paper has some drawbacks for practical application, the contribution is actually enough. Ratings can be improved further if the authors could solve my questions, especially the use of label information.

---

> ### Author Response · Authors · 2022-11-19
> **Thanks and responses to concerns**
>
> We thank you for your positive feedback and constructive suggestions, especially the paper list you kindly provided. We are pleased that you recognized the contribution of our work. Below, we address the questions raised in your comments.
>
> **1. My main concern with this paper is its usability. To analyze the complementariness, the methods need the label information beforehand. However, as we have known the label information, one could directly train the multi-modal network and percept the robustness intuitively. I think the use of label information is an obvious drawback of this work.**
>
> First, the definition of robustness requires the involvement of label information. Take adversarial robustness as an example. In a review of adversarial examples [1], the generation of an adversarial example can generally be described as a box-constrained optimization problem: Minimizes the perturbation while misclassifying the prediction with a constraint of input data. By “misclassifying the prediction,” a correct prediction exists (i.e., the label information). In a paper discussing the certified adversarial robustness [2], the evaluation of robustness also relies on the decision boundary (i.e., the label information). Similarly, the definitions of robustness in missing or noisy data cases depend on label information too.
>
> Nonetheless, our work has advantages over the direct measure of robustness. For one thing, our metric helps with the attribution of model vulnerability since high complementarity accounts a lot for model vulnerability. Instead, a direct measure of multi-modal robustness does not imply the cause of vulnerability. If we consider complementarity, the comparison of multi-modal robustness is fairer for models, and we could focus more on improvements for model and training. For another, as a reflection of the intrinsic property of multi-modal datasets and a quantification of modality interconnection, our metric can be used for analysis not limited to robustness.
>
> [1] Adversarial Examples: Attacks and Defenses for Deep Learning
>
> [2] Certified Adversarial Robustness via Randomized Smoothing
>
> **2. Some works focus on the reliability of the modality [1], which could also learn the uncertainty (importance) of each modality, which is also a kind of robustness. In contrast, the proposed method could only analyze the robustness of the datasets, ignoring an important perspective, i.e., improving the representation of the network.**
>
> We thank you for providing a related paper, and we have added the paper to our reference. This work proposed new modeling of uncertainty in multi-view learning and designed a novel method based on the Dempster-Shafer theory and the Dirichlet distribution. As a result, they mitigated the influence of the modalities with larger uncertainty on the prediction.
>
> However, their method was based on the multi-view assumption. In the paper, they assumed that each view corresponds to an evidence in the Dempster-Shafer theory and has an independent probability mass assignment. The Dempster’s combination rule they used was for independent sets of masses. In other words, each view can make a prediction independently and votes for the final prediction.
>
> Hence, in many settings of multi-modal learning, we cannot utilize their method. For instance, in Hateful-Meme Challenge, the image and the caption both have large uncertainty about the prediction. Still, their combination can have high confidence, contradictory to their rule that “when both views are of high uncertainty, the final prediction must be of low confidence.” Our method reflects this property (complementarity) which is not apparent in many multi-modal datasets.
>
> Therefore, our work has significant contributions and promotes the understanding of multi-modal datasets.

---

> > ### Author Response · Authors · 2022-11-19
> > **Follow-up**
> >
> > **3. Many recent multi-view works about information theory are missing [2-5]. All these methods attempt to establish the relationship between the different modalities and the label information in an unsupervised manner, although these methods are based on some extra assumptions, e.g., mutual redundancy [3,5] or containing view-specific information [2]. Especially, [3,4] also discuss the usability of the missing modality from information theory. The paper could do a better job by citing these relevant works and clarifying the differences between this work and them.**
> >
> > We thank you for providing related papers. We cited the paper [2] previously, and we have added other papers to the reference of the revision. The biggest difference between our work and theirs is that they all adopt the multi-view assumption. In [2] and [3], Assumption 1 restated the multi-view assumption. In [4], the objective included minimizing $H(Z_2|Z_1)$ and $H(Z_1|Z_2)$, so it discarded the complementary information. In [5], Section 3.1 stated, "This intuition relies on a basic assumption of the multi-view environment – that the two views provide the same predictive information.” This assumption does not hold in many multi-modal learning settings, as we mentioned in answer to Q2. Therefore, they did not analyze the influence of complementary information. Further, their methods could not be applied to multi-modal tasks with high complementarity.
> >
> > In [3][4], they proposed a framework for multi-view self-supervised learning based on information theory and addressed two problems: multi-view consistency and missing view recovery. In dealing with the missing data, they tried to recover them and use them for prediction, while our work theoretically analyzes the upper bound of their method, i.e., the best accuracy they can get. We demonstrated that in some datasets, it is impossible to fully recover the missing information useful for prediction.

---

> > > ### Comment · Reviewer_EDnT · 2022-11-30
> > > **Updated review**
> > >
> > > I thank the authors for their response to my concerns. The author gives further discussions on the information theory based works and robust multi-modal learning works. After reading the response, I recommend acceptance of this paper.

---

### Official Review · Reviewer_EKpK · 2022-10-21

**Confidence:** 3
**Clarity, Quality, Novelty And Reproducibility:** Good, but less novelty.
**Correctness:** 3
**Technical Novelty And Significance:** 2
**Empirical Novelty And Significance:** 2
**Recommendation:** 5

**Strength And Weaknesses:**

Strengths:
1. Although multi-modal complementariness has been focused on, this paper provides an intuitional solution (i.e. information theory) to explain the effect of this characteristic.
2. The notation and proof of this article are clear.

Weakness:
1. This work only focuses on Bayes error on a fixed model and shows how complementariness affects robustness. That is a strong assumption and thus the application scenarios are limited. What if the model takes in noisy or incomplete multi-modal data when training?
2. The experiment setting seems weak and less supportive. Through Figure 3 and Table 1, it is disturbing whether the deterioration is caused by complementariness or other nature of the dataset itself.
3. Details of these datasets had better be provided.
4. Some special meanings are best represented in the figure. For example, the meaning of $\Tau_Z$ when modality missing is quite confusing. Should it vary when data suffer from contamination?
5. This paper focused on two modalities. Then, how to define the metric in datasets with more than two modalities?
6. The word Complementarity may be more commonly used to denote complementary properties than Complementariness.


**Summary Of The Paper:**

The article investigates the robustness of multi-modal learning by introducing the concept of modality complementariness. The analysis is based on information theory, presenting the increasing Bayes error in the condition of missing and noisy modality. Then, a dataset-wise metric is proposed to measure the complementariness of each dataset. Furthermore, a dataset with more complementary information is designed in this paper. More experiments are applied to support this theory with this metric.

**Summary Of The Review:**

This paper examines the concept of Complementariness in multi-modal learning paradigm. Although the perspective is nice, some deeper insights had better to be proposed to enrich the theory.

---

> ### Author Response · Authors · 2022-11-19
> **Thanks and responses to concerns**
>
> Thank you for your detailed comments and constructive suggestions. Below, we address the concerns raised in your review.
>
> **1. This work only focuses on Bayes error on a fixed model and shows how complementariness affects robustness. That is a strong assumption and thus the application scenarios are limited. What if the model takes in noisy or incomplete multi-modal data when training?**
>
> Bayes optimal predictor is a widely-adopted assumption in analyses [1]. We use the Bayes error for analysis since we intend to focus on the relationship between modalities and the target but not the property of the hypothesis space or optimization process. The results show the upper bound of model performance under data missing/noise cases. More importantly, this analysis inspires our metric design in Section 4. As for your question, training with noisy or incomplete data is another interesting direction for future work.
>
> [1] Contrastive learning, multi-view redundancy, and linear models
>
> **2. The experiment setting seems weak and less supportive. Through Figure 3 and Table 1, it is disturbing whether the deterioration is caused by complementariness or other nature of the dataset itself.**
>
> Figure 3 shows the results of our controlled experiment. The only experimental variable is $\alpha$ for generating the synthetic dataset and $\sigma$ for generating the AAV-MNIST dataset. Other variables (e.g., hyperparameters in training and testing and dataset size) are control variables. Hence, the results demonstrate how complementarity affects robustness and verify our metric.
>
> Table 1 shows the effectiveness of our metric on real-world datasets. The deterioration in Table 1 is not only affected by complementarity. As we have mentioned in Section 5.3, there are other factors, such as the number of robust features in each modality. However, complementarity is an essential and fundamental factor among them.
>
> **3. Details of these datasets had better be provided.**
>
> Thank you for your suggestion on making our paper clearer. The descriptions of the real-world datasets are provided in the supplementary material (B.3, B.4).
>
> **4. Some special meanings are best represented in the figure. For example, the meaning of when modality missing is quite confusing. Should it vary when data suffer from contamination?**
>
> Figure 3 includes the results of three separate experiment settings (missing, noisy, and adversarial) on different datasets. We did not perform experiments where the data was both noisy and missing.
>
> The logic of the experiments on the synthetic dataset is that 1) We generated synthetic datasets with different degrees of complementarity by using different $\alpha$. 2) We calculated the metrics of these datasets and verified that our metric could quantify the complementarity as expected. 3) We tested these datasets' missing/noisy/adversarial robustness and showed the correlation between robustness and complementarity. The logic for the experiments on the AAV-MNIST dataset is the same.
>
> **5. This paper focused on two modalities. Then, how to define the metric in datasets with more than two modalities?**
>
> There are several ways to define the metric in datasets with more than two modalities.
>
> First, we can use the variant of our metric: $(\Gamma_A + \Gamma_B + … + \Gamma_C) / I(A,B,...,C; Y)$, where A, B, C represent different modalities, and $Y$ is the target. This variant resembles the total correlation in information theory used by previous work on multi-modal learning [1].
>
> Second, we can choose several modalities (e.g., A and B) among all modalities involved in the task and focus on their complementarity. In this case, we divide all modalities into two parts: \{A,B\} and \{C,...\}. So we can use another variant $\Gamma_{A,B} /  I(A,B,C,...; Y)$ for the complementarity of \{A,B\}.
>
> [1] TCGM: An Information-Theoretic Framework for Semi-Supervised Multi-Modality Learning
>
> **6. The word Complementarity may be more commonly used to denote complementary properties than Complementariness.**
>
> Thank you for your valuable suggestion. We have changed the word “complementariness” to “complementarity” in our revision, except for the title. During the rebuttal stage, the title of our paper cannot be changed.

---

> > ### Comment · Reviewer_EKpK · 2022-11-23
> > **Updated review**
> >
> > Thanks for the detailed response. Here are some I still focus on.
> > First, the theoretical analysis provides insight, but the assumptions on the Venn diagram are somewhat powerful. This will be more convincing if it can be shown in the experiment.
> > Second, the experiment results in table 1 are not very obvious, which may introduce confusion. More explanations and experiments with more than two modalities still need to be subjoined.

---

> > > ### Author Response · Authors · 2022-12-01
> > > **Thanks and responses to concerns**
> > >
> > > Thank you for your valuable feedback. Below, we address your further questions through additional explanations and experiments.
> > >
> > > > **1. The assumptions on the Venn diagram are somewhat powerful. This will be more convincing if it can be shown in the experiment.**
> > >
> > > Thank you for your valuable feedback.
> > >
> > > The Venn diagram has been commonly used to illustrate the relationship between views/modalities and the target in multi-view/multi-modal learning literature (e.g.,  Figure 1 in [1,2,3]). Specifically, TCGM[3] claimed that “conditioning on ground truth label Y, these modalities are conditionally independent” and illustrated that in their Venn diagram.  Inspired by this analysis, they proposed a surrogate goal for maximizing the mutual information of different modalities and achieved state-of-the-art performance by semi-supervised learning on some multimodal datasets.
> > >
> > > However, we notice the assumption used in [1,2,3] that the data from a single modality contains the
> > > complete label information does not apply in many other multi-modal settings.
> > > It has been mentioned in [4] and verified through experiments in previous works such as [5,6,7]. In [5], the authors observed that the uni-modal network has low accuracy in some classes (shown in Figure 1 in their paper) and verified that the uni-modal data is insufficient for predicting the correct label. In VQAv2[6] and Hateful-Meme [7], the same image paired with different text (captions or questions) can have totally different labels, so the performance of unimodal models is bounded because of incomplete information.
> > >
> > > Although previous assumptions in [1,2,3] are useful for abstraction, we argue that they are limited in understanding multimodal data in many settings. Therefore, we discard the previous multi-view assumption stating that I(X;Y|Z) and I(Z;Y|X) are negligible. Instead, we do not neglect these two quantities and focus on the two areas in the Venn diagram, which are important to understand the robustness of multimodal data. Apart from this change of focus, we do not make additional assumptions about the Venn diagram. Please let us know if you refer to any assumption other than this, and we will be glad to address your concern.
> > >
> > > **Reference**
> > >
> > > [1] Self-supervised learning from a multi-view perspective
> > >
> > > [2] Dual Contrastive Prediction for Incomplete Multi-view Representation Learning
> > >
> > > [3] TCGM: An Information-Theoretic Framework for Semi-Supervised Multi-Modality Learning
> > >
> > > [4] What Makes Multi-modal Learning Better than Single (Provably)
> > >
> > > [5] Modality Competition: What Makes Joint Training of Multi-modal Network Fail in Deep Learning? (Provably)
> > >
> > > [6] Making the V in VQA Matter: Elevating the Role of Image Understanding in Visual Question Answering
> > >
> > > [7] The Hateful Memes Challenge: Detecting Hate Speech in Multimodal Memes

---

> > > > ### Author Response · Authors · 2022-12-01
> > > > **Follow-up**
> > > >
> > > > > **2. The experiment results in table 1 are not very obvious, which may introduce confusion. More explanations and experiments with more than two modalities still need to be subjoined.**
> > > >
> > > > In table 1, in the “Our metric” column, the AAV-MNIST ($\sigma = 2.0$) and the Hateful-Meme dataset have a higher value in metric. They are observed to be more vulnerable than the other three datasets, according to the columns “Missing”, “Noisy”, and “Adversarial”. From our analysis, their complementarity contributes to their vulnerability.
> > > >
> > > > However, our pipeline may not provide an accurate value of the Modality Complementariness of some datasets due to the drawbacks of MINE(used in our pipeline).
> > > > According to a previous discussion on estimating mutual information [8], bounding mutual information in high-dimensional data remains challenging. Existing approaches, including MINE, cannot provide low-variance, low-bias estimation in some cases.  A future work direction is to lower our metric's estimation variance. For example, the estimator can be improved by substituting MINE with more efficient and accurate estimators or adjusting the network architecture of the estimator, e.g. using pre-trained big models.
> > > >
> > > > _[8] On Variational Bounds of Mutual Information_
> > > >
> > > > For experiments with more than two modalities, we add an experiment on a synthetic dataset with four modalities in the supplementary material. First, we extend our metric to cases with more than two modalities: Denote the modalities as $S=\{X_1,X_2,\dots, X_m\}$. We can choose a subset $S_1\subset S$ and focus on the complementarity of $S_1$. So we are dividing all modalities into two parts. Denote $S_2 = S\setminus S_1$, and we have the relation
> > > >        $$I(S_1,S_2;Y)  = \Gamma_{S_1} + \Gamma_{S_2} + I(S_1;Y;S_2)$$
> > > > So we define the complementarity metric of $S_1$ as
> > > >        $$\frac{\Gamma_{S_1}}{I(S;Y)} = \frac{I(S_1;Y | S_2)}{I(S;Y)}=\frac{I(S_1;Y,S_2) - I(S_1;S_2)}{I(S;Y)}$$
> > > >
> > > > We can calculate this metric using the pipeline in section 4. We test our analysis on a synthetic dataset with $m$ modalities where we adjust the degree of modality overlap by the parameter $\alpha$. Extending the settings in section 5, we generate a set of synthetic data $(\lbrace x_i|i\in [m] \rbrace, y)$ with $m=4$: First, sample random projection $P_i \in \mathbb{R}^{d_1\times d}$ and $Q_i\in \mathbb{R}^{d\times d}$, $\forall i\in [m]$ from a uniform distribution $U(-0.5,0.5)$. Then sample 10000 data points with 80/20 train/val split with the following steps:
> > > >
> > > > - Sample $x_i\in \mathbb{R}^d\sim \mathcal{N}(0,1)$, $\forall i\in [m]$.
> > > > - Set $x_i \gets (1-\alpha) x_i + \alpha x_1$ and then do projection $x_i\gets P_i x_i$, $\forall i\in [m]\setminus \lbrace 1 \rbrace$.
> > > > - Normalize all $x_i$ to unit length, $\forall i\in [m]$. If $|(x_1+x_2)\cdot (x_3+x_4)|\leq \delta$, return to the Step 2.
> > > > - Generate the label $y$: If $(x_1+x_2)\cdot (x_3+x_4) > 0$, then $y=1$; else $y=0$.
> > > > - Return the tuple $( \lbrace Q_ix_i \mid i\in [m]  \rbrace , y)$.
> > > >
> > > > We set $\langle d,d_1,d_2,\delta \rangle=\langle 50,50,50,0.25\rangle$. We calculate the metric for $S_1=\lbrace x_2 \rbrace$ and test the robustness of simple two-layer perceptron neural networks trained on the datasets with different $\alpha$: When $\alpha=0$, the modalities are independent and complementary in predicting the label $Y$. When $\alpha=1$, they are redundant for prediction. The results are shown in the following table, which shows that our pipeline can still estimate the proposed metric and quantify the complementarity of $S_1$. The missing/noisy/adversarial robustness refers to the accuracy where $x_2$ is missing/noisy/adversarially attacked, then divided by the clean accuracy. As the overlap among modalities is enlarged, the adversarial robustness increases while the other two basically hold the same. Compared with datasets with two modalities, the impact of overlap degree on the robustness towards single source attack or corruption is lighter.
> > > >
> > > >
> > > > | $\alpha$ | our metric | stdvar | clean | missing | noisy | adversarial |
> > > > | ---------------- | ------------------------- | ------------ | ------ | ----- | ----- | ----- |
> > > > | 0.0       | 1.8994 | 0.1587 | 0.8215 | 0.8405|0.7304|0.3561|
> > > > | 0.1 | 0.2459 | 0.1252 | 0.83   | 0.8470|0.7404|0.3813|
> > > > | 0.2 | 0.6623 | 0.4087 | 0.848  | 0.8550|0.7329|0.3797|
> > > > | 0.3   | -0.4451 | 0.3817 | 0.867  | 0.8806|0.7307|0.4394|
> > > > | 0.4 | -0.1692 | 0.2485 | 0.885  | 0.8712|0.7153|0.4650|

---

### Official Review · Reviewer_TcW7 · 2022-10-24

**Confidence:** 4
**Correctness:** 2
**Technical Novelty And Significance:** 2
**Empirical Novelty And Significance:** 3
**Recommendation:** 5

**Clarity, Quality, Novelty And Reproducibility:**

The background, the mathematical notations throughout the paper, and the proposed method are introduced clearly. The paper is well-written and organized in general.

**Strength And Weaknesses:**

Strengths:
1.	This paper studied a good question about what affects multimodal robustness. The conclusion is well verified and may help improve model robustness.
2.	The paper is written clearly and is easy to follow.

Weaknesses:
1.	However, my major concern is that the contribution is insufficient. In general, the authors studied the connection between the complementary and the model robustness but without further studies on how to leverage such characteristics to improve model robustness. Even though this paper could be the first work to study this connection, the conclusion could be easily and intuitively obtained, i.e., when multimodal complementary is higher, the robustness is more delicate when one of the modalities is corrupted. Except for the analysis of the connection between complementary and robustness, it is expected to see more insightful findings or possible solutions.
2.	The proposed metric is calculated on the features extracted by some pre-trained models. So the pre-trained models are necessary for metric computing which is contradictory to that the metric is used to measure the multimodal data complementary. In addition, in my opinion, the metric is unreliable since the model participates in the metric calculation and will inevitably affect the calculation results.
3.	There are many factors that will affect the model's robustness. The multimodal data complementary is one of them. However, multimodal data complementary is not solely determined by the data itself. For example, classification on MS-COCO data is obviously less complementary than VQA on MS-COCO data. As mentioned by the author, the VQA task requires both modalities for question answering, accordingly the complementary is determined by the multimodal and the target task. However, I didn't see much further discussion about these possible factors.


**Summary Of The Paper:**

This paper empirically studied the connection between the complementariness of modalities and multimodal model robustness. The authors pointed out that the multimodal models are more robust as the multimodal data are less complementary and otherwise. The conclusion is theoretically verified along with well-defined experiments. Moreover, the author proposed a new metric for measuring the complementary of multimodal data that may help improve the robustness of the multimodal model.

-------after rebbuttal
I read the resp from the authors, but only some concerns of mine have been addressed. Hence, I rise my score to 5. In fact,  if there is 4, I will only rise to it.

**Summary Of The Review:**

This paper studied the connection between the multimodal data complementary and the model robustness. The paper is well-written. However, the contribution of this paper is limited.

---

> ### Author Response · Authors · 2022-11-19
> **Thanks and responses to concerns**
>
> We thank you for your detailed summary and questions, which helped us to improve our paper. Below, we provide responses to the concerns raised in your review.
>
> **1. The contribution is insufficient. In general, the authors studied the connection between the complementary and the model robustness but without further studies on how to leverage such characteristics to improve model robustness. Even though this paper could be the first work to study this connection, the conclusion could be easily and intuitively obtained. Except for the analysis of the connection between complementary and robustness, it is expected to see more insightful findings or possible solutions.**
>
> In this paper, we study a fundamental problem of multi-modal learning: The relationship between different modalities among multi-modal data under a given task. As we have demonstrated, this relationship fundamentally determines the theoretically best performance of a multi-modal model after a single modality is missing or corrupted. Previous papers on multi-modal robustness research [1][2] do not illustrate this relationship. Instead, they treat every multi-modal dataset equally. They assume that an ideal model can predict the correct label based on only one modality. This is a strong assumption and does not hold in many multi-modal datasets. A significant contribution of our paper is providing a complete description of the relationship between different modalities and labels of multi-modal data. Although we do not directly improve the robustness of the multi-modal model, having a clear understanding of multi-modal data and modality interconnection is essential. Without this understanding, the attribution of multi-modal model vulnerability could be biased.
>
> [1] Are multimodal transformers robust to missing modality?
>
> [2] Can audio-visual integration strengthen robustness under multimodal attacks?
>
> **2. The proposed metric is calculated on the features extracted by some pre-trained models. So the pre-trained models are necessary for metric computing which is contradictory to that the metric is used to measure the multimodal data complementary. In addition, in my opinion, the metric is unreliable since the model participates in the metric calculation and will inevitably affect the calculation results.**
>
> Thanks for your insightful questions. In our paper, we use simple two-layer MLPs as the feature extractor in all the simulated datasets without pre-training on any other datasets. So we think these comparisons are relatively fair.
>
> As for natural datasets, we think there are two ways to ensure a fair comparison. First, we can use the same backbone on different datasets. In our paper, we use ResNet18 to extract visual features in Kinetics-Sound and AVE; we also use AudioNet(1-D CNN) to extract audio features in these two datasets. Second, we can leverage the fixed pre-trained big model such as Gato[1] to extract features from different modalities of various datasets, which is also a trend in the future.
>
> [1] Deepmind. A Generalist Agent.
>
> **3. There are many factors that will affect the model's robustness. The multimodal data complementary is one of them. However, multimodal data complementary is not solely determined by the data itself. For example, classification on MS-COCO data is obviously less complementary than VQA on MS-COCO data. As mentioned by the author, the VQA task requires both modalities for question answering, accordingly the complementary is determined by the multimodal and the target task. However, I didn't see much further discussion about these possible factors.**
>
> Our paper shows that complementary is co-determined by the multimodal data and the task. A multimodal dataset contains both inputs and labels. When we calculate our metric, we use the label information, which means that the task corresponding to the inputs is also critical.

---

> > ### Author Response · Authors · 2022-12-12
> > **Follow-up**
> >
> > Here we provide further response to your concerns on pre-trained models.
> >
> > First, our pipeline is designed for measuring the modality complementarity of multi-modal datasets instead of comparing the approaches used to solve multi-modal tasks. Therefore, the architecture choices and training pipelines of different approaches do not affect our metric estimation.
> >
> > Second, for the feature extraction modules in our pipeline, we suggest fixing the backbone type for each modality in different real-world datasets for a fair comparison. For instance, on Kinetics-Sound and AVE datasets, we used ResNet for visual modality and 1-D CNN for audio modality. **We do not recommend comparing metrics calculated by approaches with different feature extractors, which may induce bias.**
> >
> > With the development of multi-modal multi-task models, another option is the pre-trained multi-modal big models, such as Gato. Since we use a late-fusion style pipeline, we can easily change the feature extractors to big models. In this way, the difference among feature extractors can be eliminated.

---

### Official Review · Reviewer_L857 · 2022-10-26

**Confidence:** 3
**Correctness:** 4
**Technical Novelty And Significance:** 3
**Empirical Novelty And Significance:** 3
**Recommendation:** 6

**Clarity, Quality, Novelty And Reproducibility:**

The paper is mostly well presented. The studied problem of the interconnection between modality complementariness and multimodal robustness is novel and interesting. The calculation of the metric is reproducible.

**Strength And Weaknesses:**

Strength:

-The studied problem of how modality complementariness affects multimodal robustness is interesting. This paper provides a new angle to look at the multimodal robustness.

-The authors a thorough theoretical analysis of the interconnection of modality complementariess and robustness in multimodal setting.

-The empirical results clearly demonstrate that the proposed metric is able to quantify how complementariess of each multimodal dataset. It provides an opportunity to make a fair comparison under similar complementariness for future studies.



Questions/Weakness:

-The proposed metric provides an opportunity to make a fairer comparison of multimodal robustness among the existing methods. I’d like to see an analysis of the existing popular multimodal robustness approaches by applying the proposed metric.

-Although network architecture, training routines are out of this paper’s scope, the proposed complementariness metric is measured based on pretrained models for feature extraction. A more powerful base model usually extracts richer features. I am not clear on how to make a fair comparison if two multimodal models have different backbone feature extractors.

**Summary Of The Paper:**

This paper studies the contradiction in the existing multimodal robustness literatures. Intuitively, multimodal models are supposed to be more robust than unimodal model as extra redundancy is provided by multimodal data. However, some existing works find that multimodal integration may be more vulnerable to attack, noise or corruption. To understand this contradiction, the authors make a theoretical analysis using information theory. By formally giving the definition of complementary information, the authors establish the theoretical guarantee to quantify the different Bayes error rate of multimodal and miss-/noisy modality cases, and provide a metric to quantify the complementariness of multi-modalities. Experiments on synthetic and real world dataset show the value of measuring complementariness.

--- post-rebuttal ---\
Thanks for the authors' response. I have read the response and other reviewers' comments. Both reviewer TcW7 and I have a similar concern about the influence of pre-trained model. The authors propose two possible ways of constraining the feature extraction part to be the same. While it is still unclear to me for the case when two to-be-compared multi-modal approaches have different feature extraction modules. Despite this, I still believe there is value of being a pioneering work exploring modality complementariness, but slightly lower my score by accounting the review confidence weight.

**Summary Of The Review:**

The studied problem of how modality complementariness affects multimodal robustness is interesting. This paper provides a new angle to look at multimodal robustness. I have two general questions regarding the metric. Please see them in the weakness section.

---

> ### Author Response · Authors · 2022-11-19
> **Thanks and responses to concerns**
>
> We thank you very much for the positive feedback and appreciation of our work. We are encouraged that you found our work to be well-presented, novel, and interesting. Below, we provide responses to your questions.
>
> **1. I’d like to see an analysis of the existing popular multimodal robustness approaches by applying the proposed metric.**
>
> Thank you for your comments. We have added a section in the supplementary material (C.3) to demonstrate an application of our metric. The results show that our metric helps with the attribution of model vulnerability, while a direct measure of multi-modal robustness (i.e., accuracy in modality missing/noisy/adversarial case, which is the most popular existing multimodal robustness approach) does not. With our metric, the comparison of multi-modal robustness is fairer for models, and we could focus more on improvements for model and training.
>
> **2. A more powerful base model usually extracts richer features. I am not clear on how to make a fair comparison if two multimodal models have different backbone feature extractors.**
>
> Thanks for your insightful questions. In our paper, we use simple two-layer MLPs as the feature extractor in all the simulated datasets without pre-training on any other datasets. So we think these comparisons are relatively fair.
>
> As for real-world datasets, we believe there are two ways to ensure a fair comparison. First, we can use the same backbone on different datasets. In our paper, we use ResNet18 to extract visual features in Kinetics-Sound and AVE; we also use AudioNet(1-D CNN) to extract audio features in these two datasets. Second, we can leverage the fixed pre-trained big model such as Gato[1] to extract features from different modalities of various datasets, which is also a trend in the future.
>
> [1] Deepmind. A Generalist Agent.

---

> > ### Author Response · Authors · 2022-12-12
> > **Follow-up**
> >
> > Thank you for your feedback. Here we provide the response to your concern.
> >
> > First, our pipeline is designed for measuring the modality complementarity of multi-modal datasets instead of comparing the approaches used to solve multi-modal tasks. Therefore, the architecture choices and training pipelines of different approaches do not affect our metric estimation.
> >
> > Second, for the feature extraction modules in our pipeline, we suggest fixing the backbone type for each modality in different real-world datasets for a fair comparison. For instance, on Kinetics-Sound and AVE datasets, we used ResNet for visual modality and 1-D CNN for audio modality. **We do not recommend comparing metrics calculated by approaches with different feature extractors, which may induce bias.**
> >
> > With the development of multi-modal multi-task models, another option is the pre-trained multi-modal big models, such as Gato. Since we use a late-fusion style pipeline, we can easily change the feature extractors to big models. In this way, the difference among feature extractors can be eliminated.

---

### Decision · Program_Chairs · 2023-01-20

**Decision:**

Reject

**Justification For Why Not Higher Score:**

The contribution is relatively limited and there are still significant unresolved weaknesses.


**Justification For Why Not Lower Score:**

N/A

**Metareview: Summary, Strengths And Weaknesses:**

This paper received borderline ratings overall, with some initial divergence across reviewers. There was therefore additional discussion between the reviewers and myself after the reviewer-author discussions. In the additional discussions, there was support both for marginal acceptance and rejecting of the paper. However, reviewers had an overall lukewarm assessment of the paper, finding it to provide some value but generally limited in contribution and direct utility. The experimental validation was also considered to be weak, and there were some specific concerns about the approach such as the effect of different pre-training. Multiple reviewers stated that the author rebuttals partially addressed concerns but were not fully satisfactory. The reviewer who originally had the highest score also lowered their score. Considering the paper, and all discussions, I tend to agree that the weaknesses of the paper outweigh the contributions and do not recommend acceptance at this time.

**Summary Of Ac-Reviewer Meeting:**

Reviewers were concerned both about the limited contribution and direct utility of the paper. There are also some questions about specific aspects of the approach such as the effect of different pre-training. The author rebuttal partially but did not fully address these concerns. In the end, I concurred that the contribution of the work is somewhat limited, and in light of the various weaknesses and concerns raised that are not yet satisfied, do not recommend acceptance at this time.